# Proposal for an Organ-Specific Chronic Inflammation–Remodeling–Carcinoma Sequence

**Isao Okayasu [1,2,*], Masaaki Ichinoe [2] and Tsutomu Yoshida [2]**

[1] Division of Nutrition, Faculty of Health Care, Kiryu University, Azami 606-7, Kasakake-cho, Midori, Gunma 379-2392, Japan
[2] Department of Pathology, School of Medicine, Kitasato University, Kitasato 1-15-1, Minami-ku, Sagamihara, Kanagawa 252-0374, Japan
* Correspondence: isaokaya@gmail.com

**Abstract:** An organ-specific chronic inflammation–remodeling–carcinoma sequence has been proposed, mainly for the alimentary tract. As representative diseases, gastroesophageal reflux disease, chronic gastritis and inflammatory bowel disease (ulcerative colitis and Crohn's disease of the colitis type) were adopted for this discussion. Tissue remodeling is such an important part of tumorigenesis in this sequence that an organ-specific chronic inflammation–remodeling–carcinoma sequence has been proposed in detail. Chronic inflammation accelerates the cycle of tissue injury and regeneration; in other words, cell necrosis (or apoptosis) and proliferation result in tissue remodeling in long-standing cases of inflammation. Remodeling encompasses epithelial cell metaplasia and stromal fibrosis, and modifies epithelial–stromal cell interactions. Further, the accumulation of genetic, epigenetic and molecular changes—as well as morphologic disorganization—also occurs during tissue remodeling. The expression of mucosal tissue adapted to chronic inflammatory injury is thought to occur at an early stage. Subsequently, dysplasia and carcinoma develop on a background of remodeling due to continuous, active inflammation. Accordingly, organ-specific chronic inflammation should be ameliorated or well controlled with appropriate monitoring if complete healing is unachievable.

**Keywords:** organ-specific chronic inflammation; remodeling; metaplasia; myofibroblast; fibrosis; Barrett's esophagus; chronic atrophic gastritis; ulcerative colitis; Crohn's disease; carcinogenesis

## 1. Introduction

It is believed that chronic inflammation induces carcinoma development in various organs. This has been predominantly shown at the epidemiological level. However, the remodeling of target tissues by chronic or repeated inflammation has gone largely unnoticed and is important as a background for cancer development [1,2]. We propose a sequence of organ-specific chronic inflammation–remodeling leading to carcinoma. Typical examples include: gastroesophageal reflux disease (GERD)–Barrett's esophagus–adenocarcinoma; chronic active gastritis–atrophic gastritis–gastric carcinoma; inflammatory bowel disease (IBD), ulcerative colitis (UC) and Crohn's disease of colitis type–remodeled regenerative mucosa–dysplasia–colorectal carcinoma; chronic hepatitis–liver cirrhosis–hepatocellular carcinoma; and chronic cholecystitis–gall bladder carcinoma [3,4] in the alimentary tract. For other organs, interstitial pneumonitis–pulmonary fibrosis–lung carcinoma; and chronic human papilloma virus-infected cervicitis–uterine cervical cancer can also be added to this sequence system. Focusing on tissue remodeling induced by chronic inflammation, we summarize and discuss tumorigenesis in organ-specific chronic inflammation (Figure 1).

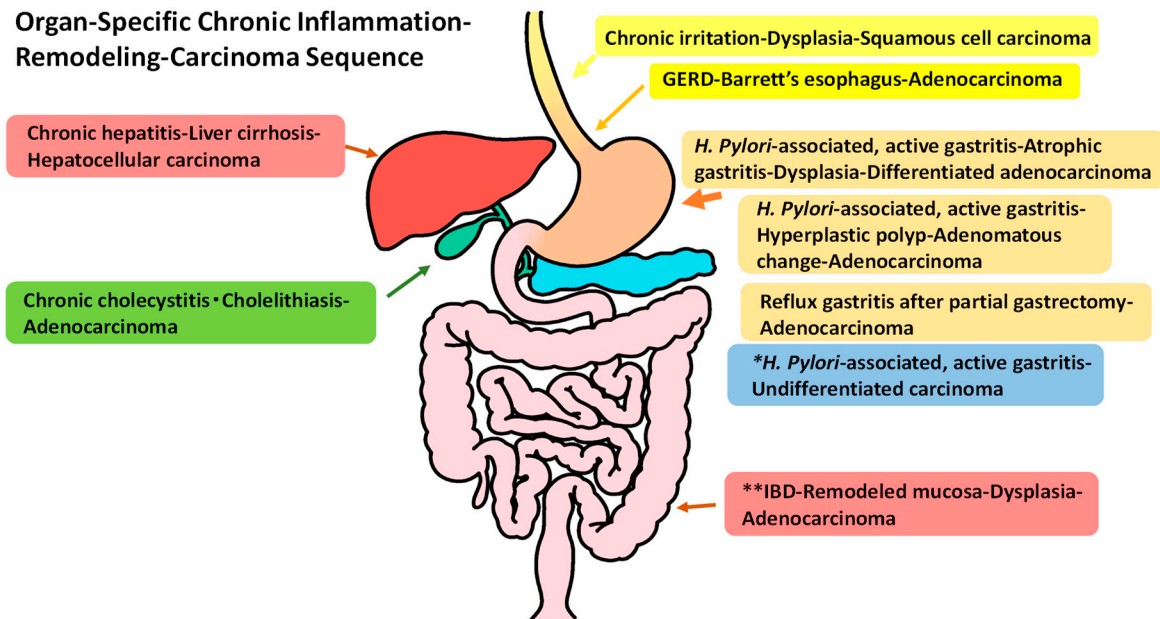

**Organ-Specific Chronic Inflammation-Remodeling-Carcinoma Sequence**

Chronic irritation-Dysplasia-Squamous cell carcinoma

GERD-Barrett's esophagus-Adenocarcinoma

Chronic hepatitis-Liver cirrhosis-Hepatocellular carcinoma

*H. Pylori*-associated, active gastritis-Atrophic gastritis-Dysplasia-Differentiated adenocarcinoma

*H. Pylori*-associated, active gastritis-Hyperplastic polyp-Adenomatous change-Adenocarcinoma

Chronic cholecystitis·Cholelithiasis-Adenocarcinoma

Reflux gastritis after partial gastrectomy-Adenocarcinoma

*\*H. Pylori*-associated, active gastritis-Undifferentiated carcinoma

\*\*IBD-Remodeled mucosa-Dysplasia-Adenocarcinoma

**Figure 1.** Typical diseases with an organ-specific chronic inflammation–remodeling–carcinoma sequence in the alimentary tract. * This sequence shows no remodeling moving toward to carcinoma development. ** inflammatory bowel disease (IBD) includes ulcerative colitis and Crohn's disease of colitis.

## 2. Organ-Specific Chronic Inflammation–Remodeling–Carcinoma Sequence

### 2.1. GERD–Barrett's Esophagus (Mucosal Remodeling)–Adenocarcinoma Sequence)

Chronic exposure of the esophageal mucosa to gastroduodenal fluid (bile salts and hydrochloric acid) causes active and erosive esophagitis, resulting in metaplasia of squamous cells to columnar epithelial cells, and submucosal fibrosis as Barrett's esophagus. It has been shown that chronic gastroesophageal reflux induces Barrett's esophagus according to clinicopathological analyses [5,6] and experimental studies using animal models [7–10]. Regarding the requirement of goblet cells (intestinal metaplasia) within the columnar-lined mucosa, the global histological definition of Barrett's esophagus is still controversial in Europe, the US, the UK and Japan [5] (Figure 2).

The risk of adenocarcinoma development in Barrett's esophagus has been established as a Barrett's esophagus–low-grade dysplasia (LGD)–high-grade dysplasia (HGD)–adenocarcinoma sequence. The risk of the development of esophageal cancer and adenocarcinoma in Barrett's esophagus is approximately 10 (10.6 relative risk) and 30 times (29.8) more, respectively, when compared with the general population [11]. The length of Barrett's esophagus is associated with a progression to HGD and adenocarcinoma [12,13]. The risk of adenocarcinoma is also greater in long segments (>6 cm) of Barrett's esophagus than in short segments (≤3 cm or >3≤6 cm) [14].

Barrett's mucosa expresses Cdx1, Cdx2 (intestine-specific transcription factor), cytokeratin 7, and villin, which are columnar epithelial markers. Fibroblasts underlying the inflammation-damaged epithelium prominently express heparin-binding epidermal growth factor-like factor (HB–EGF). The *Cdx2* promoter of squamous epithelial cells is upregulated through AP-1 and NF-kB activation via the EGF receptor (EGFR) by fibroblast-derived HB–EGF stimulation. Further, HB–EGF induces cytokeratin 7 and villin upregulation. Thus, HB–EGF released by mesenchymal cells in GERD induces the characteristic gene expression of metaplastic columnar epithelium phenotypes from esophageal squamous epithelium [15]. Acidic fibroblast growth factor 1 sequentially accumulates in Barrett's mucosa and glandular dysplasia, suggesting that it plays an important role in tumorigenesis in the esophageal mucosa [16].

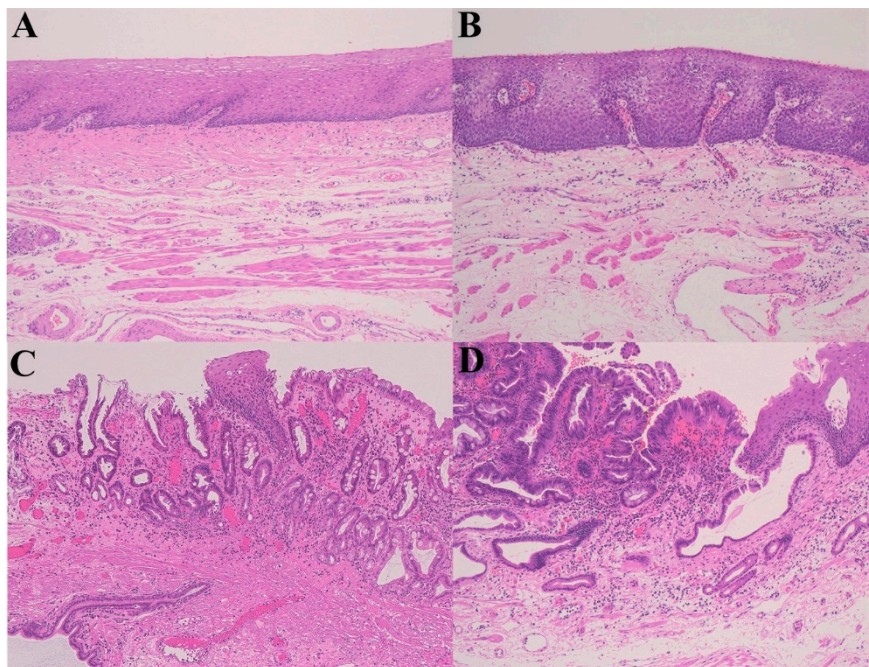

**Figure 2.** A gastroesophageal reflux disease (GERD)–Barrett's esophagus (remodeling)–high-grade dysplasia sequence. (**A**) Normal esophagus; (**B**) Gastroesophageal reflux disease (GERD); (**C**) Barrett's esophagus; (**D**) High-grade dysplasia (Hematoxylin–eosin staining, ×100).

Regarding carcinogenesis in Barrett's esophagus, various genetic and epigenetic alterations have been reported, mainly in HGD and adenocarcinoma [17]. Abnormalities of the epithelial cell-cycle occur, including apoptosis by oxidative stress and hyperproliferation due to an enhanced G1/S phase. The accumulation of mutations or overexpression of the tumor suppressor gene, *p53*, are increased in HGD and adenocarcinoma. The *p53* mutation was occasionally found even in Barrett's esophagus, suggesting that this is a critical event in carcinogenesis [18,19]. The amplification and overexpression of cyclin D [20], decreased expression and loss of heterozygosity (LOH) of suppressor genes [21], and increased expression of cyclin E [22] are also reported to be related to the progression of Barrett's esophagus to adenocarcinoma. Furthermore, characteristic DNA methylation subtypes (*ERBB2, PTPN13*) in Barrett's esophagus and adenocarcinoma were observed with regard to tumorigenesis by genome-wide DNA methylation profiling [23]. An analysis of LOH and microsatellite instability (MSI) targeting National Cancer Institute standard and tumor suppressor genes revealed that epithelial LOH was shown in both esophageal dysplasia–squamous cell carcinomas and Barrett's esophagus–adenocarcinomas. Stromal MSI occurred in a relatively early phase of a squamous cell carcinoma series. In contrast, it was relatively rare in Barrett's esophagus and an adenocarcinoma series, suggesting different pathways of tumorigenesis between the two types of esophageal carcinoma [24].

Furthermore, telomere shortening of epithelial cells has been shown in Barrett's esophagus and adenocarcinoma in relation to the degree of histological atypia, particularly with LOH-positive cases [25]. So far, in practice and according to a meta-analysis, LOH or a mutation of *p53*, and an increased level of the Ki-67 labeling index for hyperproliferation, are proposed as reliable biomarkers that predict the risk of malignant progression in Barrett's esophagus [26].

Taken together, mucosal remodeling composed of columnar cell metaplasia and mucosal fibrosis is the basis of a Barrett's esophagus–dysplasia–adenocarcinoma sequence.

The development of squamous cell carcinoma of the esophagus (ESCC) can also be included in the organ-specific chronic inflammation–remodeling–carcinoma sequence. Smoking, alcohol drinking, red and processed meat consumption, and papilloma virus infection have been put forward as risk factors for ESCC [27]. Alcohol drinking and smoking, in particular, are definite linked risk factors. A large population-based case-control study in China revealed that a significant risk for ESCC increased

in males who were heavy smokers (those who started smoking more than 20 cigarettes/day or 40 pack-years, or started smoking early) and male drinkers (those who started drinking early and with increasing duration and intensity of alcohol intake) [28]. Furthermore, on the basis of germline genotype classification of polymorphisms in *ADH1B* and *ALDH2*—which regulate alcohol metabolism—and *CYP2A6*—which is associated with DNA damage by smoking—analyses of whole-exome sequences of tumors showed that many tumors of ESCC contained mutations in genes that regulate the cell-cycle, epigenetic processes and Notch signals in Japanese patients [29]. In addition, telomeres of basal and parabasal cells were significantly shorter in the background of carcinoma in situ than in age-matched control mucosa, indicating increased chromosomal instability of nontumor mucosa [30]. Multiple lesions of dysplasia and ESCC are often found simultaneously or at different times in the same patients. ESCC may develop in any area of the esophagus, which differs from the strong tendency of a lower location in Barrett-type adenocarcinoma. Although no remarkable changes of epithelium and subepithelial stroma have been reported histologically in nontumor tissue of the esophageal wall in ESCC patients, a squamous intraepithelial neoplasia (dysplasia), low-grade, squamous intraepithelial neoplasia (dysplasia), high-grade and squamous cell carcinoma sequence is clinicopathologically used in practice [31]. Thus, remodeling at the genetic, epigenetic and molecular level is considered to occur in the background mucosa of patients with ESCC.

### 2.2. Chronic Active Gastritis–Atrophic Gastritis (Mucosal Remodeling)–Carcinoma Sequence

*Helicobacter pylori* induces active gastritis in which polymorphonuclear leukocytes infiltrate into the gastric mucosa. Persistent and active mucosal inflammation induces enhanced apoptosis and necrosis via cell-cycle arrest, by p53 activation reacting to epithelial DNA damage due to oxidative stress [32]. The regeneration of mucosal epithelia following necrosis and apoptosis accelerates in response. Regeneration causes the incomplete recovery of gastric fundic and pyloric glands, resulting in atrophic mucosa with incomplete intestinal metaplasia (colonic-type metaplasia, without Paneth cell metaplasia), as found with atrophic gastritis where both fundic and pyloric glands disappear. Mucosal remodeling includes intestinalized atrophic mucosa and a thickened muscularis mucosa with fibrosis [33]. Against this background, a differentiated-type adenocarcinoma often develops. Thus, a chronic *H. pylori*-associated gastritis–atrophic gastritis (mucosal remodeling)–differentiated adenocarcinoma (Lauren's intestinal-type) sequence is proposed (Table 1 and Figure 3).

**Table 1.** Organ-specific chronic inflammation–carcinoma (cancer) sequence in stomach.

| Chronic Inflammation | Gastritis and Remodeling | Cancer | Notes |
|---|---|---|---|
| *H. pylori*-associated gastritis | Atrophy, intestinal metaplasia Fibrosis | Differentiated type (Lauren's intestinal-type) Adenocarcinoma | Relatively elderly patients |
| *H. pylori*-associated gastritis | Active gastritis without mucosal atrophy No mucosal remodeling | Undifferentiated type (Lauren's diffuse-type) Adenocarcinoma [34–42] | Relatively young patients |
| *H. pylori*-associated gastritis | Hyperplastic polyp-adenomatous change (Dysplasia) | Adenocarcinoma [43] | *p53* mutation (41%) |
| Reflux gastritis after partial gastrectomy | Hyperplastic gastritis DNA damage, foveolar cell hyperplasia, EB virus infection | Adenocarcinoma in remnant stomach [44] | High risk after gastro-jejunostomy, enterogastric reflux |
| EB virus-associated gastritis | Atrophic gastritis, lympho-epithelioma-like histology | Adenocarcinoma [45–48] in a proximal location | Predominance among males |
| *H. pylori*-associated gastritis | Chronic severe gastritis | * MALT type lymphoma [49] | |

* Mucosa-associated lymphoid tissue (MALT) type lymphoma is not carcinoma.

Regarding another course of chronic gastritis, chronic active *H. pylori*-associated gastritis without mucosal remodeling induces an undifferentiated carcinoma (poorly differentiated adenocarcinoma; Lauren's diffuse type), which appears in relatively young patients [34]. Early diffuse-type gastric cancers include mainly poorly differentiated adenocarcinoma, pure signet ring cell carcinoma and mixed poorly differentiated adenocarcinoma. Recent clinical assessments of endoscopic submucosal dissections of early gastric cancers revealed that mixed poorly differentiated adenocarcinoma predicted endoscopic noncurative resection, compared with pure signet ring cell carcinoma [35,36], indicating a

clear difference in cancer cell differentiation and clinicopathological aspects. E-cadherin is important in maintaining cell–cell adhesion, cell survival and cell migration for tissue morphogeneis and homeostasis. Non-phosphorylated CagA of *H. pylori* binds E-cadherin or causes mutational alterations in *p53* and hyper-methylation of the E-cadherin *(CDH1)* gene, resulting in the dissociation of the E-cadherin-β-catenin complex or the impairment of E-cadherin synthesis [37]. In diffuse-type gastric cancers, mutations in or hyper-methylation of the *CDH1* gene is frequently found, suggesting disruption of E-cadherin function [38–40]. In addition, both in vitro IL-1β treatment and *Helicobacter pylori* infection induce the promoter methylation of E-cadherin in gastric cancer cell lines, indicating the activation of DNA methylation by nitric oxide production through stimulation of the NF-kB transcription [41]. Furthermore, intestinal metaplasia and intestinal-type gastric cancer also showed a relatively high frequency of alteration of the *CDH1* gene [42]. Accordingly, mutational alterations of *p53* and other factors may be associated with diffuse-type gastric cancer, as well as dysregulation of E-cadherin [37].

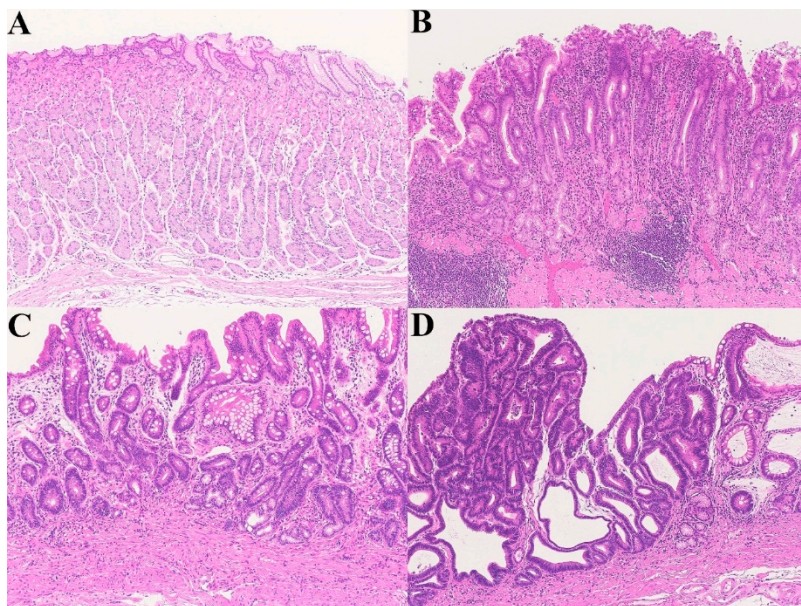

**Figure 3.** A chronic active gastritis–atrophic gastritis–high-grade dysplasia sequence. (**A**) Normal gastric mucosa; (**B**) *Helicobacter pylori*-associated, active gastritis; (**C**) Atrophic gastritis (remodeling); (**D**) High-grade dysplasia (Hematoxylin–eosin staining, ×100).

Chronic *H. pylori*-associated gastritis also induces a hyperplastic polyp–adenomatous change "dysplasia"–adenocarcinoma sequence, although it is relatively low in incidence [43]. In this type, carcinoma cells often have a *p53* gene mutation (about 41%). Adenocarcinoma in the remnant stomach develops in reflux gastritis after partial gastrectomy and gastrojejunostomy [44]. Enterogastric reflux harms gastric epithelial cells and induces DNA damage, resulting in foveolar cell hyperplasia. Adenocarcinoma on the anastomotic margin shows a high incidence of Epstein–Barr (EB) virus infection. An EB virus-associated gastritis–atrophic gastritis–adenocarcinoma sequence can also be thought of as one of several chronic gastritis–carcinoma sequences [45]. However, the frequency of EB virus-associated gastric cancer varies with region, race and era, and remains controversial. In fact, recent studies revealed that the prevalence of *H. pylori*-negative gastric cancer is very low in East Asia, representing 0.66% of gastric cancer cases in Japan [46] and 2.3% in Korea [47]. It is known that adenocarcinoma of this type is more common among males and develops in a proximal location in the stomach [45]. It is suggested that a reduction in apoptosis of EB-virus infected cells is related to carcinogenesis [48]. A mucosa-associated lymphoid tissue-type lymphoma following chronic active, *H. pylori*-associated gastritis is also described [49].

An increased risk of developing gastric carcinoma in patients with *H. pylori* gastritis has been shown in a nested case-control study with a cohort of 128,992 persons [50] and a prospective study of Japanese patients, including 1246 cases with, and 280 cases without, *H. pylori* infection [51]. Combined studies of blood samples for *H. pylori* serology and gastric cancer development in these cohorts, followed for 10 or more years, have suggested that the relative risk of gastric (non-cardia) cancer associated with *H. pylori* infection is estimated to be 5.9 times greater than for uninfected people [52]. Unsurprisingly, eradication therapy for *H. pylori* infection reduces the incidence of gastric carcinoma development [53] (Figure 1).

The course of a *H. pylori*-associated gastritis–atrophic gastritis–carcinoma sequence is long-standing. *H. pylori* infection usually occurs in childhood, mostly before 5 years of age, and varies according to country, public hygiene status, and era. In fact, *H. pylori* infections have declined recently in Japan, with improvements in economic conditions and public hygiene [54]. Active gastritis continues for a long time if the eradication of *H. pylori* is not undertaken. Cases of differentiated adenocarcinoma in atrophic gastritis are frequently seen in patients older than 60 years of age. *H. pylori* is usually not found in intestinalized atrophic mucosa. In a situation such as burned background mucosa, the presence of intestinal metaplasia of the incomplete (colonic) type is more important than that of the complete (small intestinal) type in tumorigenesis [55]. Indeed, the incidence rate of gastric adenocarcinoma was 0.72/1000 person years among 4146 patients with intestinal metaplasia that were members of Kaiser Permanente, an American-integrated managed-care consortium in Northern California [56]. At 2.56/1000 person years, the relative risk of gastric adenocarcinoma in patients with intestinal metaplasia is higher, compared with that of the Kaiser Permanente Northern California member population. The median time for gastric intestinal metaplasia to progress to adenocarcinoma was found to be 6.1 years. Another cohort study revealed that the 5-, 10- and 15-year cumulative incidences of gastric carcinoma were 0.9%, 2.0%, and 3.0%, respectively, among patients with gastric intestinal metaplasia [57]. Further, the presence of gastric background submucosal cysts was linked to early gastric cancer cases as well as gastric intestinal metaplasia and thickened muscularis mucosae, suggesting an important post-gastritis role in tumorigenesis [33]. Thus, post-gastritis–gastric intestinal metaplasia is a high-risk factor for gastric cancer.

Intestinal metaplasia is thought to be a compensatory phenomenon in response to cellular stress and misdirected differentiation. Incomplete intestinal metaplasia lacks the presence of Paneth cells at the crypt base. Such cells localize to the crypt base and secrete α-defensins that are essential for mucosal protection from intestinal microbial pathogens [58]. In comparison to complete intestinal metaplasia, incomplete intestinal metaplasia lacks these essential functions of Paneth cells, leading to an increased susceptibility to cellular stress and damage from various types of pathogens [59,60]. Expression of the homeobox gene *Cdx2* is maintained in both incomplete and complete intestinal metaplasia, and in dysplasia. SRY-related HMG-box2 (Sox2) is expressed in normal gastric mucosa, in 7% of complete intestinal metaplasia (*MUC5AC*-negative) cases and in 85% of incomplete intestinal metaplasia (*MUC5AC*-positive) cases, but only in 12% of dysplasia cases. This indicates reprogramming in gastric intestinal metaplasia and dysplasia [61]. Similarly, Cdx2 expression in incomplete intestinal metaplasia and differentiated adenocarcinoma (intestinal-type) is maintained but not so in undifferentiated carcinoma (diffuse-type), highlighting the basic role of Cdx2 in intestinal metaplasia formation and gastric carcinogenesis (intestinal-type) [62].

As for genetic and epigenetic alterations in the development of gastric cancer, an integrated profile of 55 cancer tissues using a benchtop next-generation sequencing method revealed that epigenetic alterations more frequently affected genes involved in gastric cancer-related pathways, including those of the p53 and MAPK pathways, cell-cycle regulation and oncogenes, than genetic alterations [63]. Moreover, it has been shown that DNA methylation and epigenetic silencing of the *miR-21* gene activated *STMN1* and *DIMT1*, inducing the proliferation of gastric epithelial cells inflamed with *H. pylori* [64]. In addition, the silencing of *MAP1LC3A variant 1* by methylation, essential for autophagy, was detected in 23.3% of gastric cancer mucosa and 40% of *H. pylori*-infected mucosa [65]. The analysis

of microsatellite instability (MSI) in 100 cases of sporadic gastric cancers using 10 markers of *TGFβII, IFGIIR, BAX, MSH6, EsF4, MSH2, MLH1,* and *TP53* genes showed that a high frequency of MSI (MSI-H) in gastric carcinomas was a significant predictor of antral location, intestinal type (differentiated type), *H. pylori*-seropositivity, and lower incidence of *TP53* mutations and lymph node metastasis. In contrast, gastric carcinoma with a low frequency or stable MSI showed a higher frequency of *TP53* mutations. In this study, hypermethylation of the *MLH1* promoter responsible for the loss of protein function was detected at a high frequency (13 of 14 MSI-H cancers) [66]. In another study, methylation frequency increased with disease progression by 0% (0/20 cases) in normal gastric mucosa, 28.6% (4/14) in intestinal metaplasia, 77.8% (21/27) in gastric epithelial dysplasia and 87.5% (14/16) in early gastric adenocarcinoma, according to a methylation-specific PCR analysis of five genes, including *p16, Runx3, MGMT, DAPK and RASSF1A* [67]. Thus, remodeled gastric mucosa (intestinal metaplasia) links to dysplasia and adenocarcinoma with epigenetic and genetic alterations.

Subepithelial myofibroblasts (pericryptal fibroblasts) form a subepithelial sheath and play the role of a stem cell niche at the crypt base and in the differentiation of the cryptal epithelium. The subepithelial myofibroblast sheath is maintained around crypts that consist of intestinal metaplastic epithelia but is not found in differentiated adenocarcinoma. It indicates a loss or reduction of the interaction of subepithelial myofibroblasts with mucosal epithelia in the course of the development of intestinal metaplasia to differentiated carcinoma [68]. Monocyte chemoattractant protein-1 (MCP-1) is stimulated by transforming growth factor (TGF)-β1 and is generated by α-smooth muscle actin-positive and TGF-β1 receptor-positive myofibroblasts in gastric intestinal metaplasia and gastric carcinoma. Independently of *H. pylori* coexistence, both gastric intestinal metaplasia and gastric carcinoma induce MCP-1 expression in the myofibroblasts [69]. Further, interleukin (IL)-6 production by fibroblasts is increased in the mucosal stroma of gastritis, gastric adenoma and adenocarcinoma. IL-6 is thought to activate Stat3, a transcription factor associated with gastric cancer proliferation related to cyclooxygenase-2 and inducible nitric oxide synthase activation [70]. In this situation, not only intestinal metaplasia, but also mucosal fibrosis, appears to be critical for tumorigenesis.

According to the above-described data, incomplete intestinal metaplasia (atrophic gastritis) is thought to be a component of remodeling, being a precursor of a dysplasia–carcinoma sequence.

### 2.3. IBD (Ulcerative Colitis and Crohn's Disease of the Colitis Type)–Regenerative Mucosa (Mucosal Remodeling)–Dysplasia–Carcinoma Sequence

Ulcerative colitis (UC) has characteristics of repeating active (relapsing) and inactive (remission) phases, and often subsequently develops into dysplasia and colorectal carcinoma in extended and long-standing cases (Figure 4).

The development of colorectal cancer in patients with UC is estimated to be 1.2 cases per 1000 UC patient-years, according to a meta-analysis of population-based studies from 1949 to 2011 [71]. The standardized incidence ratio (95% confidence interval [CI]) for colorectal cancer showed that it develops in more patients diagnosed with UC at a young age (8.2 cases/1000 patient-years, <30 years of age) than at a later age (1.8/1000, ≥30 years). Further, it develops in more patients with long-standing (longer than 10 years), extensive colitis (3.5 cases/1000 patient-years) than in patients without long-standing extensive colitis (0.6/1000) according to a French cohort study from 2004 to 2007 [72,73]. Such data indicate that the accumulation of inflammatory DNA damage, including oxidative stress, in the colonic mucosa induces cancer development.

Both cellular necrosis and/or apoptosis, often with mucosal erosion and submucosal ulceration, and a regenerative process of the colonic mucosa are enhanced in UC—similarly to GERD and active gastritis [74]. Repeated and continuous regeneration following chronic inflammation or repeated inflammatory damage leads to the incomplete reconstruction of the colorectal mucosa, resulting in mucosal remodeling. An increase in the immunohistochemical expression of p53 and p21, the appearance of single-stranded DNA and the Ki67 labeling index indicate DNA damage, apoptosis,

and proliferative activity, respectively, of the colonic epithelium [75]. The proteins, p16 and Bax, are upregulated in p53-wild/p53-overexpressing crypts, suggesting oxidative stress in UC [76].

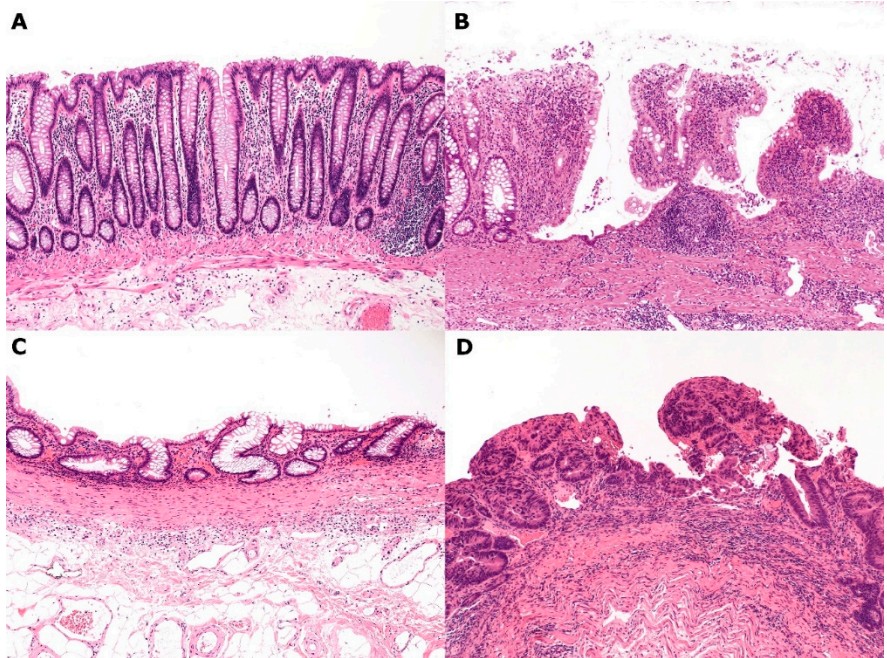

**Figure 4.** An ulcerative colitis–remodeled colonic mucosa–high-grade dysplasia sequence. (**A**) Normal colonic mucosa; (**B**) Active ulcerative colitis (UC); (**C**) Remodeled colonic mucosa in remission; (**D**) High-grade dysplasia (Hematoxylin–eosin staining, ×100).

Regarding crosstalk between inflammatory cytokines in inflammatory lesions, IL-6—produced by myeloid cells in the lamina propria and regulated by NF-kB—protects normal and inflammatory intestinal epithelia from apoptosis and shows proliferative and survival effects via mediation by Stat3 [77,78]. In UC lesions, IL-6 signaling activates tumor necrosis factor (TNF)-$\alpha$ production. TNF-$\alpha$ and IL-1$\beta$ induce the overexpression of cyclooxygenase-2 (COX-2) which leads to the production and secretion of prostaglandin E2 (PGE2) in the mucosa. Diarrhea subsequently occurs with a change in Cl- and K+ transport toward secretion in the colon [79,80]. TNF-$\alpha$ suppresses the transcription of 15-prostaglandindehydrogenase, which catabolizes PGE2, as well as the induction of COX-2 and microsomal prostaglandin E synthase-1 [81]. An experimental study using human intestinal organoids (mini-guts) showed the dual functions of COX-2–PGE2 signaling toward the differentiation of a normal epithelial lineage for mucus production, and the proliferation of annexin A1-positive inflammatory epithelium for wound closure [82]. In fact, a rectal injection of PGE2 in dextran sulfate sodium-induced rat colitis—which is usually used as a human UC model—protected against erosion and ulceration, and reduced neutrophil infiltration and myeloperoxidase activity [83]. TNF-$\alpha$ and TGF-$\beta$ also mediate oncogenic phosphorylated Smad3 [84]. Thus, inflammatory cytokines and mediators—particularly IL-6, TNF-$\alpha$, IL-1$\beta$, and PGE-2—play an important role in the inflamed mucosa.

In an overview of the genetic alterations in colorectal epithelium in UC-associated colorectal carcinogenesis as a typical organ-specific chronic inflammation–carcinoma sequence, an early event is epithelial *p53* mutation or LOH derived from cellular DNA damage—mainly caused by oxidative stress [85,86]. Subsequently, the mutation or LOH of *Rb*, the LOH of the cyclin-dependent kinase inhibitor and *p16*, mutations of the *DCC* suppressor gene and *K-ras* oncogene, and the loss of *APC* (LOH) follow in stepwise carcinogenesis. This differs from a sporadic colorectal adenoma–adenocarcinoma sequence in which the loss of APC appears early, and is followed by *K-ras*, *DCC*, and relatively late *p53* mutations [87]. Again, this difference is thought to be derived from DNA damage by oxidative stress, mainly including reactive oxygen and nitrogen species in chronic inflammation [87]. Cycles of epithelial

cell damage and regeneration lead to DNA mutation and methylation. In fact, the accumulation of mitochondrial DNA mutations was shown in both noncancerous and cancerous colorectal mucosae of UC with colitic cancer [88]. Although nuclear DNA is protected by histones, a similar mutation tendency is suggested to occur in UC.

Mutated crypts, assessed by counting a mild periodic acid–Schiff-positive, *O*-acetylation phenotype, also correlate with the duration of UC disease [89]. Crypts with mutated *p53* even appeared in regenerative colonic mucosa in long-standing UC cases and also increased in dysplasia lesions. Polyclonal *p53* gene mutations of each of the crypts were detected in regenerative crypts and LGD, indicating *p53* point mutations with heterogeneous loci. In addition, a monoclonal change in a *p53* gene mutation was found in HGD, indicating a homogenous lesion of the monoclonal *p53* gene mutation [76,90]. In comparison, the telomeres of colonic epithelia that protect important DNA in centromeres are shortened, and correlate with the duration of UC disease because of accelerated mitosis due to chronic inflammatory oxidative stress. This means increased genomic instability of the mucosal epithelia, resulting in abnormal chromosomal changes [91,92].

Regarding histological changes such as remodeling, loss, and shortness of the mucosa, and decreases in the angle of mucosal crypts, increases in fused crypts and Paneth cells, mucosal fibrosis, and a thickening of the muscularis mucosa, these correlate with the duration of UC disease [75]. Canonical discriminative analysis using immunohistochemical markers (p53, p21, Ki-67 labeling index) of remodeling, in addition to morphologic changes, can indicate a reliable risk assessment of neoplasia development [2].

Alterations of stromal tissues include a decrease of subepithelial myofibroblasts with the expression of α-smooth muscle actin, heat shock protein 47 and cytoglobin in long-standing UC [93,94]. Subepithelial myofibroblasts correspond to vitamin A–storing stellate cells and play an important role in stem cell niches at the mucosal crypt base [95]. It was experimentally suggested in a murine dextran sulfate sodium (DSS)-induced UC model that vitamin A is important in maintaining the niche function of stem cells and mucosal immunity [96,97]. Furthermore, the expression of Wnt and Notch signaling are activated at the crypt base. In contrast, bone morphogenetic protein signaling is activated and induces epithelial differentiation at the crypt top. Myofibroblasts are known to interact with intestinal epithelium to regulate the function of the epithelial stem cell niche, and induce the differentiation of epithelial cells [98,99]. Subepithelial myofibroblasts regulate epithelial cell positioning and proliferation via Wnt and Notch signaling [100]. The loss of epimorphin, a myofibroblast protein, inhibits experimental colitis and dysplasia development in murine UC and a dysplasia model developed using a combination of DSS and azoxymethane. Epimorphin is thought to have antiproliferative activity and a promorphogenetic effect in intestinal epithelium. Epimorphin deletion affects the secretion of IL-6 and regulators of the stem cell niche, such as chordin, by myofibroblasts [101]. Thus, the loss or dysfunction of subepithelial myofibroblasts means disturbed mesenchymal regulation of epithelial cells. In contrast, interstitial myofibroblasts increase with type I and type III collagen deposition [95]. Furthermore, enhanced genomic instability—including MSI and LOH—in stromal cells of regenerative mucosa without dysplasia was significantly revealed in long-standing UC cases, as well as in epithelial cells, by a combination of microdissection and the PCR–gene scan method [102,103].

Thus, mucosal remodeling of both epithelial and stromal elements, including morphologic, genomic and molecular aspects, proceeds in long-standing UC cases as the basis of neoplasia development. These characteristics clearly differ from those of sporadic colorectal carcinoma, as well as the clinicopathological [104], genetic [73,76], epigenetic [105], and molecular aspects of developed carcinomas [93,106–108].

A tendency for dysplasia and adenocarcinoma development in Crohn's disease (CD), another inflammatory bowel disease, is also known in the colorectum as well as in the small intestine. A comparative study of carcinogenesis highlighted the similarity of colorectal carcinoma in CD and UC. According to a clinicopathological analysis of 80 patients with colorectal carcinomas complicating IBD, most cancers developed after more than 8 years of disease duration (CD 75%; UC 90%) and

appeared in the area of a gross inflammatory lesion (CD 85%; UC 100%). Ages of patients diagnosed with colorectal carcinoma were similar (CD 54.5 years; UC 43.0 years) and durations of disease were long in both diseases (the median duration of disease CD 15; UC 18 years). The development of multiple carcinomas was equal in incidence (CD 11%; UC 12%). Histological features of mucinous and signet ring cell type showed an equal incidence (CD 29%; UC 21%). The simultaneous presence of dysplasia was also similar (CD 73%; UC 100%). Overall 5-year survival rates were almost equal (CD 46%; UC 50%) [109]. A meta-analysis revealed that the overall relative risk of colorectal cancer was 4.5 in patients with the CD of colorectal disease, in comparison with 1.1 in the CD of ileal disease [110]. Furthermore, a clinicopathological analysis of 313 surgically treated patients with CD of colitis found 11 adenocarcinomas (3.5%); multivariate analysis revealed disease duration (>10 years), age at CD diagnosis, distal localization, and penetrating disease as significant risk factors [111]. Thus, colitic carcinomas develop in the CD of colitis, similarly to UC. Although genetic and molecular research has not been performed, mucosal remodeling of regenerated epithelium and stromal fibrosis can be seen in the background of developed dysplasia and carcinoma. Accordingly, an organ-specific chronic inflammation–remodeling–carcinoma sequence can be applied to long-term Crohn's disease of a colitis type.

## 3. Conclusions

An organ-specific chronic inflammation–remodeling–carcinoma sequence is a plausible theory to explain the development of carcinoma after chronic inflammation. Epithelial cell metaplasia—extended regeneration—and mucosal and submucosal fibrosis, are common elements in the remodeling of this sequence in gastrointestinal diseases, including GERD, chronic gastritis, UC, and CD of the colitis type. Remodeling involves a background of genetic, epigenetic and molecular alterations as well as morphological abnormalities, the accumulation of which allows the promotion of dysplasia and carcinoma. Disorganized interactions between epithelial and stromal cells is also thought to be the key to remodeled mucosa developing into a carcinoma. To prevent this sequence, chronic active inflammation should be adequately treated and controlled following an accurate diagnosis. Appropriate monitoring of inflammation should also occur, such as endoscopic examinations, to evaluate the mucosa in combination with other supplementary tests that include reflux monitoring in GERD [112,113], *H. pylori* detection tests in *H. pylori*-associated gastritis [114,115], and the measurement of prostaglandin E major urinary metabolites in UC [116].

**Author Contributions:** Conceptualization, I.O.; Methodology, M.I., T.Y.; Writing the Original Draft, I.O.; Writing, Reviewing & Editing, I.O., M.I., T.Y.

**Funding:** This research received no external funding.

**Acknowledgments:** The authors thank Miss M. Iwasaki for her excellent schematic drawing.

**Conflicts of Interest:** The authors declare no conflicts of interest.

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
