# Peer review of "Proposal for an Organ-Specific Chronic Inflammation–Remodeling–Carcinoma Sequence"

_gastrointestdisord, doi:10.3390/gidisord1030028_

Round 1

Reviewer 1 Report

This review is very simple and didatic. There are no novel informations.

Author Response

This review is very simple and didatic. There are no novel informations.

Answer. Thank you for your comment.

As the reviewer has written, our proposal is quite simple. But, from an overall viewpoint, we want to highlight organ-specific chronic inflammation–remodeling–carcinoma sequences for gastroenterologists. As we have described in this review, the remodeling consists of mucosal fibrosis due to chronic inflammation as well as epithelial cell metaplasia or regenerated disorganization. This remodeling includes different kinds of molecular changes involved in genetic and epigenetic alterations toward tumorigenesis. We believe that this review will be helpful for gastroenterologists in their practice of medicine.

We would like to ask you to read our revised manuscript, again.

Reviewer 2 Report

It is a very insightful and well-written review. It is also tutorial. 

The author focus on tissue remodeling as an important process to carcinogenesis, but carcinogenesis include an irreversible point such as fixed mutation caused by covalent bond adduction of chemical carcinogen to DNA Base (which finally lead to mis-paring and mutation). The address of these lines would further give the readers better perspective in this field.

Author Response

Thank you for your adequate suggestion.

We have added one paragraph with references (page 8, line 280).

In an overview of the genetic alterations in colorectal epithelium in UC-associated colorectal carcinogenesis as a typical organ-specific chronic inflammation–carcinoma sequence, an early event is epithelial p53 mutation or LOH derived from cellular DNA damage mainly caused by oxidative stress [75, 76]. Subsequently, the mutation or LOH of Rb, the LOH of the cyclin-dependent kinase inhibitor and p16, mutations of the DCC suppressor gene and K-ras oncogene, and the loss of APC (LOH) follow in stepwise carcinogenesis. This differs from a sporadic colorectal adenoma–adenocarcinoma sequence in which the loss of APC appears early, and is followed by K-ras, DCC, and relatively late p53 mutations [77]. Again, this difference is thought to be derived from DNA damage by oxidative stress, mainly including reactive oxygen and nitrogen species in chronic inflammation [77]. Cycles of epithelial cell damage and regeneration lead to DNA mutation and methylation. In fact, the accumulation of mitochondrial DNA mutations was shown in both noncancerous and cancerous colorectal mucosae of UC with colitic cancer [78]. Although nuclear DNA is protected by histones, a similar mutation tendency is suggested to occur in UC.

Reviewer 3 Report

This is a review article on organ-specific chronic inflammation–carcinoma sequence. Among several organs, the authors described only about Barrett’s adenocarcinoma, differentiated-type (Lauren’s intestinal-type) gastric cancer, and colitic cancer from UC. Although it is well-written, there are few points to be checked as follow.

1. To strengthen the manuscript, please add descriptions on 

(1) undifferentiated-type (Lauren’s diffuse-type) gastric cancer induced by active H. pylori infection, 

(2) marginal cancer after gastrectomy which occurs at the anastomosis site owing to long-term, reflux-related injury, and 

(3) adenocarcinoma arising from gastric hyperplastic polyp. 

A table showing different remodeling & carcinogenesis process in comparison with differentiated-type gastric cancer, would be helpful.

2. In similar, please add different inflammation-carcinoma process between Barrett’s adenocarcinoma and esophageal squamous cell carcinoma. 

Also, add differences between colitic cancers from UC and those from large intestinal-type, Crohn’s disease. 

Please focus more on the prostaglandin-cytokine crosstalk in inflammation including the role of tumor necrosis factor.

Round 2

Reviewer 1 Report

Dear Authors, the paper is well write, but I did find new aspect or original criticism. I think this is not particularly  of interest for the readears. It is a good resume of the literature, the literature is not always the more recent papers.

Author Response

We appreciate your comments.

1) Along the line of suggestions, we adopted recent references so far as we could.

References #5-10; #35-37; #40-42; #46-47: #112,113, 115.

2) One sentence was added with one reference, as follows. Line 54-56, page 2.

 Regarding the requirement of goblet cells (intestinal metaplasia) within the columnar-lined mucosa, the global histological definition of Barrett’s esophagus is still controversial among Europe/US and UK/Japan [5].

Reviewer 3 Report

Thank you for answering to the comments. Although most are revised properly, there are misleading messages as follow.

1. With regard to the undifferentiated type gastric cancer, please quote recent references rather than using old references published before 2000. Scirrhous carcinoma is no longer considered as equal to diffuse-type gastric cancer. Most of the undifferentiated gastric cancers are "non-scirrhous type" which can be resected even by endoscopic submucosal dissection [Ansari S et al. Int J Mol Sci 2018;19(8),  Kwak DS et al. Gut Liver 2018;12:263-70,  Horiuchi Y et al. Gastric cancer 2016;19"515-23]. 

2. It is no longer considered that E-cadherin methylation related only to diffuse-type gastric cancer. It is induced by Hp infection, and therefore, E-cadherin gene mutation is found both in intestinal-type and diffuse-type gastric cancers (Huang FY et al. Cancer 2012;118:4969-80,  Chan AO et al. Gut 2003;52:502-6). In countries where Hp infection with East-Asian type cagA is common, promoter methylation of E-cadherin is found both in differentiated-type and undifferentiated-type gastric cancers. 

3. In the last part of Table 1, it is summarized that "10% of all gastric carcinomas are linked to EBV-associated gastritis" based on old reference. Recently, it is known that most of the gastric cancers origin from Hp infection. In Japan and Korea, only 0.66% and 2.3% of gastric cancers origin from causes other than Hp infection (Matsuo T et al. Helicobacter 2011;16:415-9,  Kwak HW et al. J Gastroenterol Hepatol 2014;29:1671-7). In East Asian countries, EBV-related gastric cancer is very rare.

Author Response

Thank you for your comments.

We appreciate your adequate suggestion.

#1. The term “scirrhous carcinoma” was our careless mistake.

We have deleted the term “scirrhous carcinoma”,

We have added some explanation of diffuse-type gastric carcinoma. Line 143-151, page 5.

Early diffuse-type gastric cancers include mainly poorly differentiated adenocarcinoma, pure signet ring cell carcinoma and mixed poorly differentiated adenocarcinoma. Recent clinical assessments of endoscopic submucosal dissections of early gastric cancers revealed that mixed poorly differentiated adenocarcinoma predicted endoscopic noncurative resection, compared with pure signet ring cell carcinoma [35, 36], indicating a clear difference in cancer cell differentiation and clinicopathological aspects.

#2. We have added discussion regarding E-cadherin with recent references. Line 151-162, page 5.Table 1 was also changed.

E-cadherin is important in maintaining cell-cell adhesion, cell survival and cell migration for tissue morphogeneis and homeostasis. Non-phosphorylated CagA of H. pylori binds E-cadherin or causes mutational alterations in p53 and hyper-methylation of the E-cadherin (CDH1) gene, resulting in dissociation of the E-cadherin-β-catenin complex or impairment of E-cadherin synthesis [37]. In diffuse-type gastric cancers, mutations in or hyper-methylation of the CDH1 gene is frequently found suggesting disruption of E-cadherin function [38-40]. In addition, both in vitro IL-1β treatment and Helicobacter pylori infection induce promoter methylation of E-cadherin in gastric cancer cell lines, indicating the activation of DNA methylation by nitric oxide production through stimulation of NF-kB transcription [41]. Furthermore, intestinal metaplasia and intestinal-type gastric cancer also showed a relatively high frequency of alteration of the CDH1 gene [42]. Accordingly, mutational alterations of p53 and other factors may associate with diffuse-type gastric cancer as well as dysregulation of E-cadherin [37].

#3. Regarding EB virus-associated gastric cancer, we have added some explanation with recent references. Line 170-174, page 5, 6.    Table 1 was also changed.                                                                                          An EB virus–associated gastritis–atrophic gastritis–adenocarcinoma sequence can also be thought of as one of several chronic gastritis–carcinoma sequences [45]. However, the frequency of EB virus-associated gastric cancer varies with region, race and era, and remains controversial. In fact, recent studies revealed that the prevalence of H. pylori-negative gastric cancer is very low in East Asia, representing 0.66% of gastric cancer cases in Japan [46] and 2.3% in Korea[47].  

Round 3

Reviewer 1 Report

bibliography is improved